# Seven Weeks of High-Dose Vitamin D Treatment Reduces the Need for Infliximab Dose-Escalation and Decreases Inflammatory Markers in Crohn’s Disease during One-Year Follow-Up

**DOI:** 10.3390/nu13041083

**Published:** 2021-03-26

**Authors:** Mia Bendix, Anders Dige, Søren Peter Jørgensen, Jens Frederik Dahlerup, Bo Martin Bibby, Bent Deleuran, Jørgen Agnholt

**Affiliations:** 1Department of Hepatology and Gastroenterology, Aarhus University Hospital, 8200 Aarhus, Denmark; andedige@rm.dk (A.D.); soerjoer@rm.dk (S.P.J.); jensdahl@rm.dk (J.F.D.); joeragnh@rm.dk (J.A.); 2Medical Department, Randers Regional Hospital, 8930 Randers, Denmark; 3Department of Public Health—Department of Biostatistics, Aarhus University, 8000 Aarhus, Denmark; bibby@ph.au.dk; 4Department of Rheumatology, Aarhus University Hospital, 8200 Aarhus, Denmark; bd@biomed.au.dk; 5Department of Biomedicine, Aarhus University, 8000 Aarhus, Denmark

**Keywords:** vitamin D treatment, Crohn’s disease, infliximab

## Abstract

Background: Seven weeks of high-dose vitamin D treatment decreases intestinal IL17A and IFN-γ mRNA expression in active Crohn’s disease (CD). In this follow-up study, we investigated whether seven-week vitamin D treatment affected the infliximab response in the following 45 weeks compared to placebo. Methods: CD patients (*n* = 40) were initially randomised into four groups: infliximab + vitamin-D; infliximab + placebo-vitamin-D; placebo-infliximab + vitamin-D; and placebo-infliximab + placebo-vitamin-D. Infliximab (5 mg/kg) or placebo-infliximab was administered at weeks 0, 2 and 6. Vitamin D (5 mg bolus followed by 0.5 mg/day for 7 weeks) or placebo-vitamin D was handed out. After the 7-week vitamin D period, all patients received infliximab during follow-up. Results are reported for Group D+ (infliximab + vitamin-D and placebo-infliximab + vitamin-D) and Group D- (infliximab + placebo-vitamin-D and placebo-infliximab + placebo-vitamin-D). Results: Group D- patients had greater needs for infliximab dose escalation during follow-up compared to group D+ (*p* = 0.05). Group D+ had lower median calprotectin levels week 15 (*p* = 0.02) and week 23 (*p* = 0.04) compared to group D-. Throughout follow-up, group D+ had 2.2 times (95% CI: 1.1–4.3) (*p* = 0.02) lower median CRP levels compared with group D-. Conclusions: Seven weeks high-dose vitamin D treatment reduces the need for later infliximab dose-escalation and reduces inflammatory markers. EudraCT no. 2013-000971-34.

## 1. Introduction

Crohn’s disease (CD) is characterised by chronic intestinal inflammation elicited by a dysregulated immune response towards the commensal microbiota. Immune modulation and inhibition are cornerstones in the treatment strategy, which often includes biological therapy. The dominating biological therapy used is anti-tumour necrosis factor alpha (anti-TNFα) antibody treatment. Other biological therapies used in CD comprise anti-interleukin (IL) 12 and 23 antibody treatment (Ustekinumab) and anti-integrin alpha4 beta7 antibody treatment (Vedolizumab) [1,2].

Anti-TNFα treatment induce remission in two third of the treated CD patients. However, the effects of anti-TNFα antibody treatment decreases over time, and about 25% of the anti-TNFα antibody treated patients loses response to treatment within the first year [3,4]. This increases the risk of inflammation, hospitalization and need for surgery [5]. Therefore, additional treatments, which could increase or maintain the efficacy of the anti-TNFα antibody treatment response, are of great importance.

Vitamin D is a prosteroid and lipophilic hormone, mainly produced in skin cells by photolysis of 7-dehydrocholesterol to vitamin D3 by ultraviolet spectrum of sunlight. Vitamin D_3_ is hydroxylated twice (25-hydroxylation by the liver and 1-hydroxylation mainly by the kidney) to become biological active 1,25-dihydrovitamin D3 (1,25-vitD) [6]. Active vitamin D_3_ binds to the nuclear -vitamin D receptor (VDR) [7], which is expressed in most tissues and cells. The 1.25-vitD-receptor complex binds to vitamin D responding elements and function as a nuclear transcription factor [8]. Vitamin D is involved in the calcium homeostasis and immune functions. In CD, vitamin D is an immune modulator that change the adaptive immune response, the mucosal barrier and the microbiome [9].

In CD, vitamin D treatment has been demonstrated to decrease the risk of flare-ups [10,11]. A recent metanalysis concluded that vitamin D treatment improved disease activity and decreased the inflammatory marker C-reactive protein (CRP) in CD patients [12]. Other studies have shown an association between vitamin D deficiency and increased disease activity in CD [13,14]. Reduced efficacy of anti-TNFα antibody treatment has also been associated with vitamin D deficiency in IBD [15,16]. In infliximab treated IBD patients, vitamin D levels were correlated with infliximab trough plasma concentrations and both parameters were associated with remission [17].

In a recent randomized trial we showed that seven weeks of high dose vitamin D treatment to CD patients with disease activity decreased the mucosal mRNA expression of the proinflammatory cytokines IL17A and IFN-γ [18]. Both cytokines are believed to be central in CD inflammation [19]. The present study investigates whether initial high-dose vitamin D treatment improved the following clinical response to 45 weeks of infliximab treatment, i.e., reduced the need for dose-escalation. Furthermore, we investigated if high dose vitamin D improved the anti-inflammatory effects of infliximab.

## 2. Materials and Methods

### 2.1. Study Approval and Conduction

The study was a single-centre study carried out at Aarhus University Hospital, Denmark from July 2014 to October 2017. The study was approved by the Danish Medicine Agency (EudraCT no. 3013-000971-34), the Central Denmark Regional Committee for Health Research Ethics (no. 1-10-72-141-13) and the Danish Data Protection Agency (no. 1-16-02-296-13). The study was conducted according to the guidelines of good clinical practice (GCP) and was monitored by the GCP unit at Aarhus University Hospital. Upon completion of the study, inclusion data were transferred into Research Electronic Data Capture (REDCap) [20].

### 2.2. Former Intervention Study

The present study is an observational follow-up study. The former intervention study from week 0 to seven is detailed described in Bendix et al. [18]. In brief 40 CD patients with disease activity were randomised double blinded into one of four groups:Infliximab and vitamin D_3_ (Ifx + VitD);Infliximab and placebo vitamin D_3_ (Ifx + placeboVitD);Placebo infliximab and vitamin D_3_ (placeboIfx + VitD);Placebo infliximab and placebo vitamin D_3_ (placeboIfx + placeboVitD).

CD patients receiving vitamin D_3_ were treated with an oral bolus of 200,000 international units (IU) (5 mg) Dekristol^®^ followed by an oral daily dose of 20,000 IU (0.5 mg) for seven weeks. Patients in the placebo vitamin D_3_ group received placebo capsules.

Patients randomized to infliximab received infliximab infusion with 5 mg/kg at baseline, week 2 and week 6. Patients randomized to placebo infliximab were treated with 0.9% sodium chloride infusions at the same timepoints.

Patients were examined with colonoscopy week 0 and 7 including evaluation with the Crohn’s Disease Endoscopic Index of Severity (CDEIS). If patients experienced significant clinical progression of disease activity a rescue endoscopy was performed instead of the week seven endoscopy and the project treatment was unblinded. These patients received infliximab 5 mg/kg as rescue treatment (placebo infliximab treated patients) or 10 mg/kg infliximab (if initial treatment was infliximab 5 mg/kg). Rescue treated patients were included in follow-up.

Patients with endoscopic improvement at week seven continued with open infliximab treatment with first infusion week 15. Patients with unchanged or increasing CDEIS scores week seven were unblinded regarding infusion treatment. Patients who had received placebo infliximab treatment were treated with open (induction) infliximab 5 mg/kg (week 8, 10 and 14) followed by every 8 weeks. Patients who had been treated with infliximab treatment received an extra bolus 10 mg/kg infliximab and afterwards 5 mg/kg every 8 weeks. For further details we refer to the intervention study [18].

### 2.3. Follow-Up

Patients were included in the follow-up if they had completed the intervention period including the two endoscopies and hereby continued open infliximab treatment. Treatment failures were defined as patients with need for abdominal surgery and discontinuation of infliximab treatment due to loss of response or side effects. These patients were excluded from further follow-up from the time of the ceased infliximab treatment.

During follow-up the patients were sampled into two groups (Figure 1):Group D+ (containing the Ifx+VitD and placIfx+vitD groups)Group D- (containing the Ifx+placVitD and placIfx+placVitD groups)

After the initial 7 weeks of intervention patients were followed for 45 weeks in follow-up (a total of 52 weeks from baseline). Patients had the following follow-up visits calculated from baseline: week 15 (+/− one week), week 23 (+/− one week), week 31(+/− one week); and week 52 (+/− four weeks). Patients received infliximab infusions at all follow-up visits. Harvey Bradshaw Index Crohn´s Disease Activity (HBI) [21] was measured the day of each infliximab infusion. CRP and albumin were measured within +/− 14 days of visits. Faecal calprotectin was measured within +/− 30 days of each visit. Per standard treatment protocol, azathioprine was added to infliximab to reduce the risk of anti-TNF alpha antibodies development between week 15 and 23 in those not receiving azathioprine or methotrexate at baseline.

### 2.4. Vitamin D Safety Markers

Serum 25-OH vitamin D_2_ + D_3_ was analysed using high-performance liquid chromatography-tandem mass spectrometry on mass spectrometers API3000 TM or API5500 TM (Applied Biosystems, Lincoln, OR, USA). 1.25-dihydroxyvitamin D_3_ was measured using liquid chromatography mass spectrometry and detected by Electrospray Ionization Tandem Mass Spectrometry. The 5500 Triple Quadrupole or 6500 QTRAP (AB Sciex, Framingham, MA, USA) were used for these analyses. Serum ionised calcium was measured by a potentiometric method on a Nova CRT 8 electrolyte analyser (Diamond Diagnostics Inc., Holliston, MA, USA). Plasma PTH was measured by an immunometric assay and plasma phosphate by absorption photometry, both on a Cobas^®^ 6000 (Roche Diagnostics, IN, USA).

### 2.5. Statistical Analyses

The repeated measurement data were analyzed using a linear mixed-effects regression model with initial infliximab (yes/no), vitamin D (yes/no) and corresponding follow-up visit along with the interaction between them as categorical fixed effects. Patients were included as a random effect. An unstructured error variance-covariance matrix was chosen to allow for possible difference in correlations and standard deviations between measurements corresponding to different visits. After inspection of plots of standardized residuals versus fitted values and QQ-plots of the standardized residuals, analysis was performed on all measurements using a logarithmic scale. Results are given as estimated medians (back-transformed means on the logarithmic scale) with 95%-confidence intervals (95% CI) for combinations of vitamin D and follow-up visit as there was little insignificant effect of initial infliximab. Groups are compared using ratios of estimated medians (back-transformed mean differences) with 95% CI. Tests for no interaction between vitamin D and follow-up visit (parallel log-mean curves) were performed and in the case of acceptance, an overall median ratio is reported with 95% CI. The repeated measurements data were analyzed using Stata version 16.1. Relative risks and Fisher’s exact test were calculated using Prism 6 (GraphPad Software Inc., La Jolla, CA, USA). 

## 3. Results

### 3.1. Treatment Failures and Need for Dose-Escalation during Follow-Up

Group D+ and Group D- consisted of, respectively, 22 and 13 patients week 15 (Figure 1). At this time point group D+ had in average received 3.0 infliximab dosages while group D- had received 3.5 dosages in average (*p* = 0.21). We tested if receiving infliximab from week 0 affected the faecal calprotectin and CRP levels during follow-up compared to starting infliximab week 7. There was no effect of receiving infliximab from week 0 compared to infliximab from week 7 (faecal calprotectin: *p* = 0.87, CRP: *p* = 0.91).

The need for dose escalation of infliximab was assessed at every visit and is showed in Table 1. No patients in group D+ demonstrated treatment failure or need for infliximab dose escalation week 15–31. In group D- five patients needed infliximab dose-escalation week 15–31 (*n* = 1 week 15, *n =* 2 week 23, *n* = 2 week 31) and one patient demonstrated treatment failure. From week 32 to 52 three patients in Group D+ needed infliximab dose-escalation and three patients demonstrated treatment failure (1 loss of response, 2 side effects). In this period one patient in Group D- needed infliximab dose escalation and two patients in dropped out (1 moved to another region of the country, one in remission did not consent for further treatment). Overall, infliximab dose-escalation was necessary in 3 out of 22 Group D+ patients (14%, 95% CI: 3–35%) and six out of 13 (46%, 95% CI: 19–75%) group D- patients (relative risk: 0.33, 95% CI: 0.09–0.99) (*p* = 0.05) (Table 1).

The observed difference in need for infliximab dose-escalation was not dependent on patients’ use of azathioprine. At week 23, 82% of group D+ and 100% of group D- were treated with azathioprine (*p* = 0.27).

### 3.2. Initial High-Dose Vitamin D Treatment Reduces Faecal Calprotectin and CRP Levels

Patients who had received seven weeks of high-dose vitamin D treatment (group D+) during intervention, had lower faecal calprotectin levels in follow-up at weeks 15 and 23 compared to the placebo-vitamin D treated group (group D-). Faecal calprotectin median levels were below 70 mg/kg in group D+ at both weeks 15 (66 mg/kg, 95%, CI: 35–124 mg/kg) and 23 (68 mg/kg, 95%, CI: 33–138 mg/kg). Though, in group D- faecal calprotectin median levels were 3.8 times higher at both weeks 15 (249 mg/kg, 95%, CI: 108–576 mg/kg) (*p* = 0.02) and weeks 23 (261 mg/kg, 95%, CI: 100–682 mg/kg) (*p* = 0.04) (Figure 2A). In group D- where five out of 13 had been infliximab dose-escalated week 15 to 31, faecal calprotectin levels decreased week 31. From weeks 31 to 52, faecal calprotectin levels in group D- decreased to calprotectin levels comparable with group D+.

Plasma CRP levels were within the normal range, but group D- had a 2.2 times higher median curve CRP level through all 52 weeks of follow-up compared with group D+ (95%, CI: 1.1–4.3) (*p* = 0.03) (Figure 2B). HBI and albumin levels did not differ significantly between the groups (data not shown).

### 3.3. Increased Vitamin D Levels Are Maintained for 24 Weeks after 7 Weeks High-Dose Treatment

Seven weeks of vitamin D treatment did increase levels of 25-hydroxyvitamin D (25-vitD) and 1.25-dihydroxyvitamin D (1.25-vitD) during follow-up as expected. Both 25-vitD and 1.25-vitD levels depended on treatment (*p* < 0.0001) (Figure 3A,B). 25-vitD levels stayed elevated in group D+ compared to group D- until week 31. Here, 25-vitD levels in group D+ had decreased to 25-vitD levels comparable to groups D-. During follow-up high dose vitamin D treatment remained safe. No patients experienced hypercalcemia or hyperphosphatemia and both calcium-ion and phosphate levels stayed within reference levels through follow-up (data not shown).

## 4. Discussion

In the present study, we investigated if seven weeks of high-dose vitamin D treatment to CD patients with disease activity reduced need for infliximab dose-escalation and decreased the inflammatory response during 45 weeks of continued infliximab treatment. CD patients who had received seven weeks with high-dose vitamin D had less need for infliximab dose escalation during follow-up compared to patients receiving placebo. In addition, vitamin D treatment reduced faecal calprotectin and CRP levels during follow-up compared to placebo.

During weeks 15 and 23, group D+ had lower faecal calprotectin levels compared to group D-. Group D- had continued elevated median faecal calprotectin levels above 250 mg/kg, while group D+ had levels below 70 mg/kg. Faecal calprotectin levels above 200 mg/kg are associated with mucosal inflammation in CD [22,23,24]. Our finding of reduced faecal calprotectin and CRP levels from the addition of high-dose vitamin D to infliximab therapy indicates that high-dose vitamin D may enhance the mucosal anti-inflammatory effects of infliximab. This is supported by a recent clinical trial investigating CD patients with moderate disease activity and ongoing infliximab treatment. These patients were treated in 8 weeks with different vitamin D dosages weekly. Only the highest dosage of 50000 IU/week resulted in reduced faecal calprotectin levels [25]. This indicates that high and supra-physiological dosages of vitamin D are needed to obtain a disease modifying effect. Whether the effects of high-dose vitamin D are additive or synergistic to infliximab is unknown and not possible to decipher from our study. However, other studies supporting the present results demonstrated that in anti-TNFα treated CD patients high 25-vitD levels and anti-TNFα trough-concentrations were associated with mucosal healing [26]. Mucosal healing is perceived an important treatment goal in CD [27]. Another IBD study demonstrated that vitamin D deficiency was associated with disease activity and loss of clinical response to biological treatment [28].

During weeks 15 to 31 five out of 13 patients in Group D- needed infliximab dose escalation, which could explain the observed decrease in faecal calprotectin levels in this group at week 31.

The anti-inflammatory effects of seven weeks high-dose vitamin D treatment were maintained until week 31. At week 31 the group D+ 25-vitamin D levels were no longer increased. In addition, dose escalation of infliximab was also needed in Group D+ between weeks 31 to 52. Further studies are needed to investigate if ongoing vitamin D treatment together with infliximab could maintain the increased treatment response.

Our results support that the addition of high-dose vitamin D to infliximab treatment modulates mucosal immune activity, thereby contributing to the achievement and maintenance of remission. During active inflammation in CD vitamin D did not significantly reduce the disease activity, but decreased the mucosal IL17A and Interferon gamma expression [18].

How high-dose vitamin D treatment modulates the different immune functions is beyond the scope of this study. However, several studies have suggested opposite immune function modulations between vitamin D and TNFα. Vitamin D receptor (VDR) expression in intestinal epithelial cells (IEC) decreases IEC apoptosis and thereby inhibits experimental colitis [29]. In active IBD inflammation, TNF-α down-regulates the VDR expression on IEC. This has been shown to increase inflammation-induced IEC apoptosis in experimental colitis [30]. Hence, the effects of high-dose vitamin D in CD may be reduction of TNF-α mediated IEC apoptosis and hereby enhancement of the mucosal barrier function [31,32]. In addition, vitamin D treatment has demonstrated an anti-inflammatory shift in the intestinal bacterial composition in CD patients and in experimental colitis mice [33,34].

Our study does have limitations. This study reports the follow-up period of our original intervention study where 40 patients were randomized into four groups receiving either infliximab or placebo-infliximab and high-dose vitamin D or placebo vitamin D. In the present study, the original four groups were sampled into two groups, which might entail a risk for different numbers of infliximab infusions in Group D+ and Group D- before entering the reported follow-up period. Group D- had a received an insignificant higher average of infliximab dosage at week 15 than group D+. This did not influence the faecal calprotectin and CRP levels in the follow-up period.

The sample size was small which increased the risk of type two errors. However, the initial vitamin D or placebo treatment remained blinded throughout follow-up. To avoid selection bias, we combined data from rescue-treated patients (week three) with week seven data from patients who followed the full protocol. The patients who underwent rescue treatment had only received two infusions and three weeks of vitamin D/placebo treatment, which might influence the results.

## 5. Conclusions

In this observational follow-up study CD patients initially treated with high-dose vitamin D had a reduced need for infliximab dose escalation and decreased faecal calprotectin and CRP levels during on-going infliximab treatment for 45 weeks. Present study indicates that high-dose vitamin D may reduce the need for infliximab dose-escalation. Hence, adding high-dose vitamin D may be used to maintain patients´ response to infliximab treatment. However, a larger blinded placebo-controlled study is needed to confirm our results and to elaborate which minimum dosage of vitamin D is needed.

## Figures and Tables

**Figure 1 nutrients-13-01083-f001:**
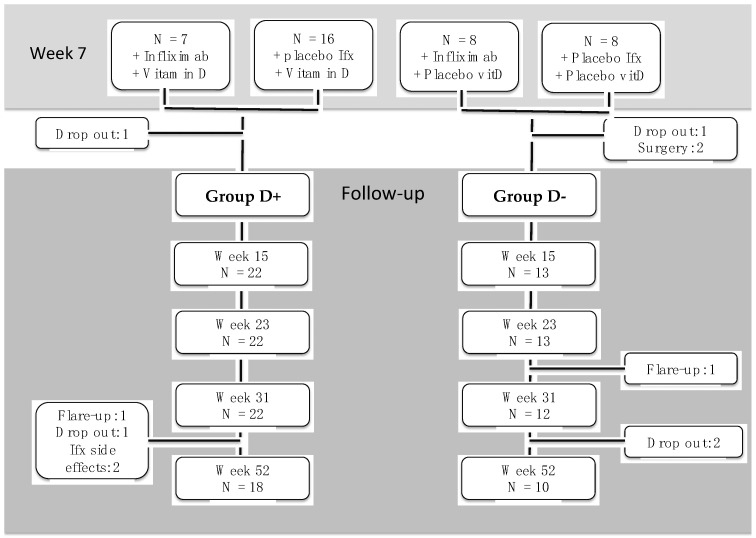
Patient flow from end of intervention study to follow-up week 52. Patients were excluded when discontinuing infliximab treatment. Ifx; infliximab, vitD; vitamin D, N; number.

**Figure 2 nutrients-13-01083-f002:**
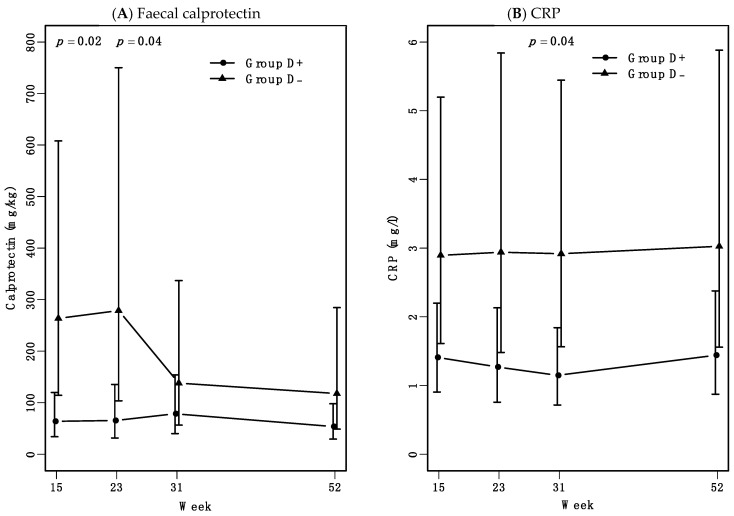
Faecal calprotectin and CRP levels from weeks 15 to 52. During follow-up all patients were treated with infliximab. Data are presented as estimated medians with 95% confidence interval (CI). (**A**) Faecal calprotectin levels in Group D- were 3.8 times higher at both week 15 and week 23 compared to group D+. At weeks 31 and 52, calprotectin levels were not significantly different between the two groups. The overall median curves of the two groups tended to be non-parallel (mixed model, *p* = 0.1). (**B**) Median curves for CRP levels over time were close to parallel, but shifted. Group D- had a 2.2 times higher median curve compared with group D+ (*p* = 0.04).

**Figure 3 nutrients-13-01083-f003:**
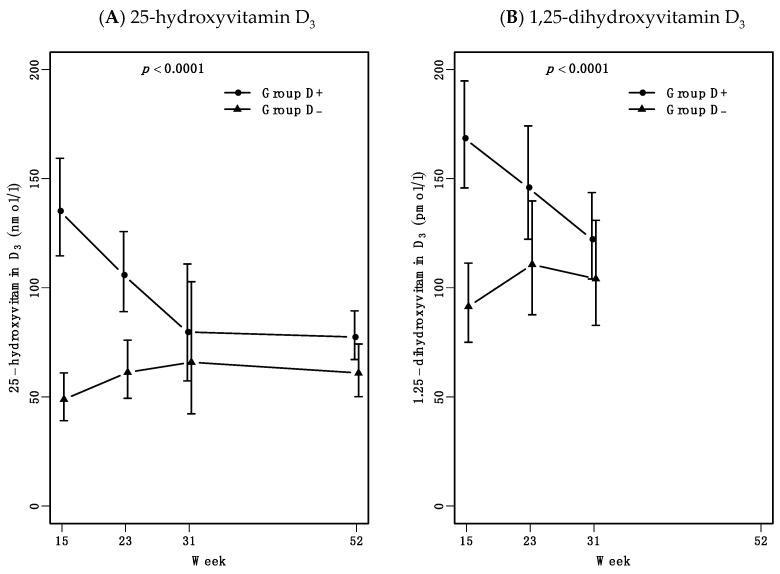
25-hydroxyvitamin D and 1.25-dihydroxyvitamin D levels during follow-up. Results are presented as estimated medians with 95% confidence interval (CI), p value represent mixed model test for parallel curves. (**A**). 25-hydroxyvitamin D_3_ levels were dependent of vitamin D treatment as expected. At week 31 group D+ had decreased to comparable levels with group D-. (**B**). Active 1.25-dihydroxyvitamin D_3_ levels week 15 to 31 were also depended of vitamin D treatment.

**Table 1 nutrients-13-01083-t001:** Infliximab treatment and dose escalation during follow-up.

	Group D+	Group D-	*p*-Value
*Patients in treatment groups, n*	22	13	
Relapse and infliximab stopped, *n*	1 (5%)	1 (8%)	0.7
Infliximab dose escalation, *n* *	3 (14%)	6 (46%)	0.05
Infliximab dose escalation week 15, *n*	0	1 (8%)	0.4
Infliximab dose escalation week 23, *n*	0	2 (15%)	0.1
Infliximab dose escalation week 31, *n*	0	2 (15%)	0.1
Infliximab dose escalation week 32–52, *n*	3 (14%)	1 (8%)	1.0
Oral prednisolone 50 mg added week 15	0	1 (8%)	0.4
Stop infliximab patients in remission (patient wish), *n*	1 (5%)	2 (15%)	0.3
Stop infliximab due to side effects, *n*	2 (9%)	0	0.5
Azathioprine users week 23, *n*	18 (82%)	13 (100%)	0.3

Table 1. Patients were followed until week 52 or if they stopped infliximab treatment. Differences in groups are tested with Fisher’s exact test. * Shortened infliximab interval was defined as treatment intervals shortened from week 8 to weeks 7, 6 or 4 due to increased disease activity.

## Data Availability

Results, protocol information with more are available on www.clinicaltrialsregister.eu/ctr-search/trial/2013-000971-34/DK.

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
