# Peer review of "Seven Weeks of High-Dose Vitamin D Treatment Reduces the Need for Infliximab Dose-Escalation and Decreases Inflammatory Markers in Crohn’s Disease during One-Year Follow-Up"

_nutrients, 2021, doi:10.3390/nu13041083_

Round 1

Reviewer 1 Report

An original study on the use of high dose vitamin do to reduce the needs for infliximab in Crohn's affected patients; The study gives a lot of interesting information; one of the main limitations is in my opinion the low number of patients, as a bigger study would have probably reached higher levels of statistical significance. I have some queries:

The introduction paragraph should in my opinion be enlarged; you should expand and at least cite the possible biologic treatments of CD.

page 1 line34-37 "Crohn’s disease (CD) is characterized by chronic intestinal inflammation elicited by a dysregulated immune response towards the commensal microbiota. Immune modulation and inhibition are cornerstones in the treatment strategy, which often includes biological therapy like anti-tumor necrosis factor-alpha (anti-TNF) antibody treatment."

Also, the conclusions should in my opinion be expanded; what the results of this study suggest to the clinician? Is high-dose vitamin D in your opinion a treatment that could help in the management of CD treated with infliximab? Are other studies necessary to confirm the results? please also expand this paragraph

Thank You

Author Response

Dear reviewer 1 

Thank you for your competent suggestions.

1. We have extended the introduction according to reviewers´ suggestions and have added information about possible biological treatments in Crohn’s disease at page 1:

The dominating biological therapy used is anti-tumour necrosis factor alpha (anti-TNFa) antibody treatment. Other biological therapies used in CD comprise anti-interleukin (IL) 12 and 23 antibody treatment (Ustekinumab) and anti-integrin alpha4 beta7 antibody treatment (Vedolizumab)[1, 2]

Anti-TNFatreatment induce remission in two third of the treated CD patients.

2. We have extended the conclusion (page 9) with the following:

” Present study indicates that high dose vitamin D may reduce the need for infliximab dose-escalation. Hence, adding high-dose vitamin D may be used to maintain patients´ response to infliximab treatment. However, a larger blinded placebo-controlled study is needed to confirm our results and to elaborate which minimum dosage of vitamin D is needed. “ 

Reviewer 2 Report

A well-structured original article on vitamin D supplementation in the management of infliximab treated Crohn disease; Only minor queries:

In the introduction, a more detailed general description of vitamin D is in my opinion necessary.

The conclusions must be expanded, as they are too short.

No need to report non-significant p values in table 1 

Author Response

Dear reviewer 2

Thank you for suggestions, they will improve the quality of our paper. We have revised the paper with the following:

1) More detailed general description of vitamin D

The introduction (page 2) have been extended with general details about vitamin D as suggested:

Vitamin D is a prosteroid and lipophilic hormone, mainly produced in skin cells by photolysis of 7-dehydrocholesterol to vitamin D3 by ultraviolet spectrum of sunlight. Vitamin D3is hydroxylated twice (25-hydroxylation by the liver and 1-hydroxylation mainly by the kidney) to become biological active 1,25-dihydrovitamin D3 (1,25-vitD) [6]. Active vitamin D3binds to the nuclear -vitamin D receptor (VDR) [7], which is expressed in most tissues and cells. The 1.25-vitD-receptor complex binds to vitamin D responding elements and function as a nuclear transcription factor[8]. Vitamin Dis involved in the calcium homeostasis and immune functions.In CD, vitamin D is an immune modulator that change the adaptive immune response, the mucosal barrier and the microbiome [9].”

2) Need for expansion of conclusions

The conclusions have been extended as suggested with the following (page 9):

” Present study indicates that high dose vitamin D may reduce the need for infliximab dose-escalation. Hence, adding high-dose vitamin D may be used to maintain patients´s response to infliximab treatment. However, a larger blinded placebo-controlled study is needed to confirm our results and to elaborate which minimum dosage of vitamin D is needed. “ 

3. The reviewer suggests that nonsignificant p-values are removed in table 1.  We agree that the significant differences are the most interesting results, but we do believe that different nonsignificant p-values add information for the reader if two groups are comparable or tended to be different. This information could be important for the reader’s understanding of the results. Therefor we have kept the nonsignificant p-values in table 1.

Reviewer 3 Report

Dear authors, thanks for this interesting short paper dealing with a Crohn's disease. The presentation of your experiment and the results are quite good, but in my opinion, your article could be improve by presenting some hypothesis trying to explain the results of your experiment. Could you present some hypothesis linked with the vit D physiological activity and the mechanism of action of Infliximab ? The results on calprotectin and CRP are of interest and may help to elaborate hypothesis. Could toy please develop ? It will improve the quality of your article.

Author Response

Dear reviewer 3

Thank you for your excellent suggestion, we have added the following hypothesis which might explain the vitamin D physiological activity and the mechanism of action of Infliximab (page 8):

1)  Inclusions of hypothesis regarding vit D physiological activity and infliximab action

“How high-dose vitamin D treatment modulates the different immune functions is beyond the scope of this study. However, several studies have suggested opposite immune function modulations between vitamin D and TNFa. Vitamin D receptor (VDR) expression in intestinal epithelial cells (IEC) decreases IEC apoptosis and thereby inhibits experimental colitis[29]. In active IBD inflammation, TNF-adown-regulates the VDR expression on IEC. This has been shown to increase inflammation-induced IEC apoptosis in experimental colitis[30]. Hence, the effects of high-dose vitamin D in CD may be reduction of TNF-α mediated IEC apoptosis and hereby enhancement of the mucosal barrier function[31, 32]. In addition, vitamin D treatment has demonstrated a anti-inflammatory shift in the intestinal bacterial composition in CD patients and in experimental colitis mice[33, 34]. “

Round 2

Reviewer 1 Report

The authors responded to all queries. The article is eligible to be published